# Acrometastases to the Hand: A Systematic Review

**DOI:** 10.3390/medicina57090950

**Published:** 2021-09-09

**Authors:** Giuseppe Emmanuele Umana, Gianluca Scalia, Paolo Palmisciano, Maurizio Passanisi, Valerio Da Ros, Gianluca Pompili, Fabio Barone, Paolo Amico, Santino Ottavio Tomasi, Francesca Graziano, Iolanda Valeria Patti, Stefania Mele, Rosario Maugeri, Giovanni Raffa, Giuseppe Roberto Giammalva, Gerardo Domenico Iacopino, Antonino Germanò, Giovanni Federico Nicoletti, Massimo Ippolito, Maria Gabriella Sabini, Salvatore Cicero, Lidia Strigari, Giacomo Cuttone

**Affiliations:** 1Trauma Center, Gamma Knife Center, Department of Neurosurgery, Cannizzaro Hospital, 95126 Catania, Italy; paolo.palmisciano94@gmail.com (P.P.); mpassanisi@tiscali.it (M.P.); fbarone1969@gmail.com (F.B.); cicerosalvatore@yahoo.it (S.C.); 2Department of Neurosurgery, Highly Specialized Hospital and of National Importance “Garibaldi”, 95122 Catania, Italy; gianluca.scalia@outlook.it (G.S.); fragraziano9@gmail.com (F.G.); gfnicoletti@alice.it (G.F.N.); 3Department of Biomedicine and Prevention, University Hospital of Rome “Tor Vergata”, 00133 Rome, Italy; darosvalerio@gmail.com; 4Plastic Surgery Unit, Cannizzaro Hospital, 95126 Catania, Italy; gianluca.pompili@yahoo.it; 5Department of Pathological Anatomy, Cannizzaro Hospital, 95126 Catania, Italy; p.amico80@gmail.com; 6Department of Neurological Surgery, Christian Doppler Klinik Paracelsus Medical University, 5020 Salzburg, Austria; s.tomasi@salk.at; 7Laboratory for Microsurgical Neuroanatomy, Christian Doppler Klinik, 5020 Salzburg, Austria; 8Medical Physics Unit, Cannizzaro Hospital, 95126 Catania, Italy; valeriapatti72@gmail.com (I.V.P.); mele.stefania@gmail.com (S.M.); mgabsabini@gmail.com (M.G.S.); 9INFN-Laboratori Nazionali del Sud, Via S. Sofia 62, 95123 Catania, Italy; cuttone@lns.infn.it; 10Post-Graduate Residency Programme in Neurological Surgery, Department of Experimental Biomedicine and Clinical Neurosciences, School of Medicine, Neurosurgical Clinic, AOUP “Paolo Giaccone”, 90127 Palermo, Italy; rosario.maugeri1977@gmail.com (R.M.); robertogiammalva@gmail.com (G.R.G.); gerardo.iacopino@gmail.com (G.D.I.); 11BIOMORF Department, Division of Neurosurgery, University of Messina, 98124 Messina, Italy; giovanni.raffa@unime.it (G.R.); germano@unime.it (A.G.); 12Department of Advanced Technologies, Nuclear Medicine and PET Cannizzaro Hospital, 95126 Catania, Italy; ippolitomas@yahoo.it; 13Department of Medical Physics, IRCCS University Hospital of Bologna, 40138 Bologna, Italy; lidia.strigari@aosp.bo.it

**Keywords:** acrometastases, thumb metastases, carcinoma, immunotherapy, chemotherapy, radiotherapy

## Abstract

*Background and Objectives:* The term acrometastases (AM) refers to secondary lesions sited distally to the elbow and knee, representing 0.1% of all bony metastases. By frequency, pulmonary cancer and gastrointestinal and genitourinary tract neoplasms are the most responsible for the reported AM. Improvements in oncologic patient care favor an increase in the incidence of such rare cases. We performed a systematic review of acrometastases to the hand to provide further insight into the management of these fragile patients. We also present a peculiar case of simultaneous acrometastasis to the ring finger and pathological vertebral fracture. *Material and Methods:* A literature search according to the PRISMA (Preferred Reporting Items for Systematic Reviews and Meta-Analyses) statement was conducted using the PubMed, Google Scholar, and Scopus databases in December 2020 on metastasis to the hand and wrist, from 1986 to 2020. MeSH terms included acrometastasis, carpal metastasis, hand metastasis, finger metastasis, phalangeal metastasis, and wrist metastasis. *Results:* In total, 215 studies reporting the follow-up of 247 patients were analyzed, with a median age of 62 years (range 10–91 years). Overall, 162 out of 247 patients were males (65.6%) and 85 were females (34.4%). The median reported follow-up was 5 months (range 0.5–39). The median time from primary tumor diagnosis to acrometastasis was 24 months (range 0.7–156). Acrometastases were located at the finger/phalanx (68.4%), carpal (14.2%), metacarpal (14.2%), or other sites (3.2%). The primary tumors were pulmonary in 91 patients (36.8%). The average interval from primary tumor diagnosis to acrometastasis varied according to the primary tumor type from 2 months (in patients with mesenchymal tumors) to 64.0 months (in patients with breast cancer). *Conclusions:* Acrometastases usually develop in the late stage of oncologic disease and are associated with short life expectancy. Their occurrence can no longer be considered rare; physicians should thus be updated on their surgical management and their impact on prognosis and survival.

## 1. Introduction

The term acrometastases (AM) refers to secondary lesions sited distally to the elbow and knee [1], which represent 0.1% of all bone metastases [2]. Usually, AM develop during follow-up of oncologic patients, but in 10% of cases, their occurrence is the first sign of a previously undiagnosed tumor [3]. This rare finding is usually detected in multi-organ neoplastic patients, thus indicating a short life expectancy (6 months) after their diagnosis. By frequency, pulmonary cancer and gastrointestinal and genitourinary tract neoplasms are the most responsible for the reported AM [4]. Improvements in oncologic patient care favor an increase in such rare cases [5]. Macroscopically, acrometastases show as painful, swollen, and reddened lesions that affect soft tissue and cause a reduced range of motion. Due to their characteristics, these lesions need to be distinguished from primary integumentary neoplasm, pyogenic granuloma, osteomyelitis, tuberculosis, inflammatory processes, cysts, gout, and ganglia [2,6,7,8,9,10,11,12,13,14,15,16,17,18,19,20,21,22,23,24,25,26,27,28,29,30,31]. A systematic review was performed on acrometastases to the hand and wrist to provide further insights into the management of these fragile patients, who have an increased risk of developing acrometastases due to improvements in treatment, and the subsequent extension of life expectancy.

## 2. Materials and Methods

### 2.1. Study Selection

A literature search on metastases to the hand and wrist was performed in accordance with the PRISMA (Preferred Reporting Items for Systematic Reviews and Meta-Analyses) guidelines in December 2020 using the PubMed, Google Scholar, and Scopus databases, from 1986 to 2020, and registered to PROSPERO (ID 277733) [32]. Relevant MeSH terms included acrometastasis, carpal metastasis, hand metastasis, finger metastasis, phalangeal metastasis, and wrist metastasis. This search was restricted to case reports, case series, letters to the Editor, and abstracts. The reference lists of all articles and cross-references were checked and, if relevant, were included in this systematic review. Additionally, non-English-language papers were considered relevant and included in this study after translating the relevant data. The PRISMA flow diagram for our search is outlined in Figure 1.

### 2.2. Data Extraction

A total of 254 articles were identified, but 39 were excluded because they had redundancies in their publications, concerned foot acrometastases, or presented ambiguous, repeated, or incomplete data. This operation yielded 215 articles with a total of 247 patients, who were enrolled in our systematic review. The articles were written in English, French, German, Hungarian, Italian, Spanish, Portuguese, Russian, Turkish, and Japanese. All included studies were meticulously reviewed and scrutinized for their study design, methodology, patient characteristics, and the following data points: authors; year of publication; and patients’ age, sex, metastatic locations, primary tumor origin, interval time from primary tumor diagnosis to acrometastasis treatment options, and survival (months) when available. The extracted data are presented in Appendix A [3,4,6,7,8,9,10,11,12,13,14,15,16,17,18,19,20,21,22,23,24,25,26,27,28,29,30,31,33,34,35,36,37,38,39,40,41,42,43,44,45,46,47,48,49,50,51,52,53,54,55,56,57,58,59,60,61,62,63,64,65,66,67,68,69,70,71,72,73,74,75,76,77,78,79,80,81,82,83,84,85,86,87,88,89,90,91,92,93,94,95,96,97,98,99,100,101,102,103,104,105,106,107,108,109,110,111,112,113,114,115,116,117,118,119,120,121,122,123,124,125,126,127,128,129,130,131,132,133,134,135,136,137,138,139,140,141,142,143,144,145,146,147,148,149,150,151,152,153,154,155,156,157,158,159,160,161,162,163,164,165,166,167,168,169,170,171,172,173,174,175,176,177,178,179,180,181,182,183,184,185,186,187,188,189,190,191,192,193,194,195,196,197,198,199,200,201,202,203,204,205,206,207,208,209,210,211,212,213,214,215,216,217].

## 3. Results

In total, 215 studies reporting the follow-up of 247 patients were analyzed. The median age was 62 years (range 10–91 years). Overall, 162 out of 247 (65.6%) patients were males and 85 (34.4%) were females. The median (range) reported follow-up was 5 (0.5–39) months. The median time from primary tumor diagnosis to acrometastasis was 24 months (range 0.7–156). Acrometastases were located at the finger/phalanx (68.4%), carpal (14.2%), metacarpal (14.2%), or other sites (3.2%). The primary tumors were pulmonary in 91 (36.8%), gastrointestinal in 62 (25.1%), urinary tract tumors in 33 (13.4%), neck tumors in 17 (6.9%), breast tumors in 10 (4.0%), pelvic tumors in 8 (3.2%), skin tumors in 5 (2.0%), bone tumors in 3 (1.2%), blood tumors in 3 (1.2%), mesenchymal tumors in 3 (1.2%) of unknown origin in 3 (1.2%), and other types in 9 (3.6%) patients. The average interval from primary tumor diagnosis to acrometastasis varied according to the primary tumor type from 2 months (in patients with mesenchymal tumor) to 64.0 months (in patients with breast cancer) (Table 1). The distribution of acrometastases’ localization versus the type of primary site is shown in Figure 2.

## 4. Discussion

### 4.1. Pathophysiology

Bone metastases show tropism towards red marrow, found mainly in vertebral bodies. Since hands contain little red marrow, they are rarely a site of bone metastasis [146,218,219]. Moreover, bone metastases usually affect bones with low blood flow within the capillaries, and this also explains why acrometastases are so rare as such a blood perfusion setting is scarce in distal bones [147,220,221,222,223]. AM’s development and pathophysiology are not yet fully understood. The most plausible hypothesis is that malignant cells metastasize to distal bones through blood, not lymphatic circulation [1,148]. The most frequent primitive tumor-related cause of AM is lung cancer, likely because of the connection with the heart and blood circulation, bypassing the pulmonary and liver filter. AM can be a secondary localization of several other primitive cancers, from esophageal squamous cell carcinoma [150] to gastric adenocarcinoma [169] following orthopedic surgery for a pathologic fracture [155]. In this systematic review of the literature, 215 studies reported the outcome of 247 patients with multi-organ concurrent metastases, mainly from lung cancer in 91 (36.8%), gastrointestinal cancer in 62 (25.1%), urinary tract cancer in 33 (13.4%), neck cancer in 17 (6.9%), breast cancer in 10 (4.0%), pelvic cancer in 8 (3.2%), skin cancer in 5 (2.0%), bone cancer in 3 (1.2%), blood cancer in 3 (1.2%), mesenchymal cancer in 3 (1.2%), an unknown origin in 3 (1.2%), and other types in 9 (3.6%) patients. The finger/phalanx (68.4%), carpal (14.2%), and metacarpal (14.2%) sites are the prevalent sites of AM because chemotactic factors released after continued traumatisms stimulate the release of prostaglandins, which favor tumor cell adhesion [152,218]. Furthermore, the distal phalanges have higher blood perfusion than the proximal phalanges [153]. The importance of blood supply and chemotaxis has been further postulated after a report of AM from colon adenocarcinoma following a dog bite wound [154]. Direct seeding has been suggested after a patient, affected by lung cancer metastasis and who used to digitally occlude his tracheostomy to speak better, developed thumb metastasis [155]. Voskuil et al. [142] reported a patient with history of colon adenocarcinoma one year before who complained of worsening wrist pain after trauma (car wheel change). He underwent a carpectomy and survived one year after AM diagnosis. Other authors have reported direct or indirect traumatisms before the development of AM, confirming the pathophysiological hypothesis suggesting that trauma plays an important role [128,129,130].

### 4.2. Imaging

Osteolytic areas are detected by standard hand X-ray (Figure 3); AM from breast cancer can appear as sclerotic lesions [4,224], while metastases from prostatic cancer have osteoblastic features [3,148]. To better identify suspicious radiological findings at full-body examination, isotope bone scans have proven useful [3,56,148,225]. Positron emission tomography (PET) represents an advanced and comprehensive diagnostic tool that allows a precise differential diagnosis of benign and malignant tumors, detecting primary and secondary tumors before other diagnostic tools with 90% sensitivity and 78% specificity [157,226] (Figure 4). A case of an asymptomatic metastatic lesion to the fourth metacarpal from vaginal squamous cell carcinoma, discovered during a PET examination at follow-up, has been reported in the literature [158]. CT scans offer low resolution at the level of small structures. Magnetic resonance imaging (MRI) is an accurate diagnostic methodology to characterize metastatic bone and related soft tissue involvement [6].

### 4.3. Patient Population

From the systematic analysis of the literature, the age at presentation of AM was found to progressively increase, with a median age of 62 years (range 10–91 years), probably due to improvements in therapeutic management, even if the overall survival remains unaltered [160,220,227]. The incidence of AM is greater in men because primary lung cancer, which is the main cause of these metastases, is more frequent in the male population [3,146,227]. No association has been documented between right and left hands or dominant and non-dominant hands, although a greater incidence has been reported in the dominant hand due to more frequent traumatisms [3,146,227]. Lechmiannandan et al. [139] reported a patient affected by proximal humerus metastasis from clear cell renal carcinoma who developed AM to his right thumb one month after orthopedic surgery. The AM were treated surgically one month later, with progressive worsening; the patient was referred for palliative treatment. This experience suggests a higher risk of AM following orthopedic surgery for pathological fractures.

### 4.4. Therapeutic Management

AM develop in the late stage of oncologic disease, often characterized by multi-organ spread of secondary tumors. Thus, the management of those patients depends on the comprehensive evaluation of the clinical condition, and the therapeutic strategy is defined on a patient-specific basis, without established guidelines [51,228,229,230]. The main goals of the treatment are pain control and function sparing of the hand of interest in the setting of a demolitive surgery represented by phalanx or finger amputation [3,146,227]. Life expectancy is influenced by the primary tumor [218], but the development of AM represents progression of the disease to distal districts and is thus associated to the very last period of survival of the patient. AM developed after a mean of 2 months in patients affected by mesenchymal tumors but after a mean of 64 months in those affected by breast cancer. These data suggest that patients suffering from highly malignant tumors may develop earlier AM compared to those dealing with less malignant neoplasms. In 30% of the studies reported in the literature, AM can be the first presentation of diffuse tumor. When present, the association of the acral swelling together with the pathological sacral fracture may make the diagnosis easier, allowing to provide prompt treatment and comprehensive oncologic management. Distal acral tumefaction associated to ulceration, pain, and progressive worsening should prompt evaluation for possible metastasis, especially in oncologic patients but also in apparently healthy patients. AM have traditionally been considered a rare finding, but reports of them in recent years have increased, probably thanks to improvements in the therapeutic protocols of oncologic patients, and it is reasonable that they will further increase in the future. Hayden et al. [218] suggested that the real incidence of AM could be much higher considering the risk of unreported cases due to misdiagnosis.

## 5. Conclusions

Acrometastases usually develop in the late stage of oncologic disease and are associated with poor life expectancy. Their occurrence can no longer be considered rare. According to this review on acrometastases, which is the first systematic report to date, AM should be considered a finding that may be faced in daily practice. Physicians should be updated on AM, their surgical management, and their impact on prognosis and survival.

## Figures and Tables

**Figure 1 medicina-57-00950-f001:**
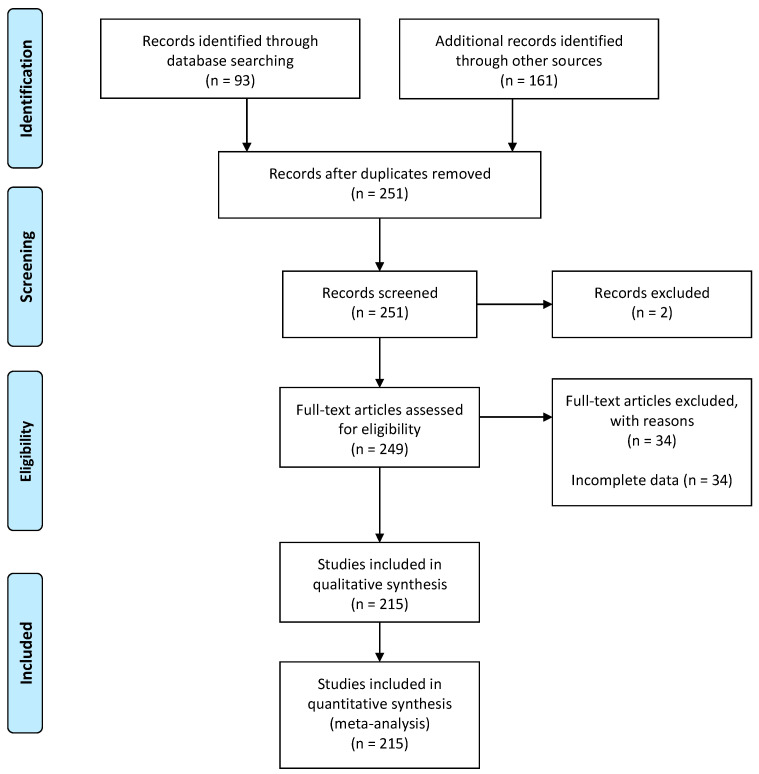
PRISMA flow diagram regarding acrometastases to the hand.

**Figure 2 medicina-57-00950-f002:**
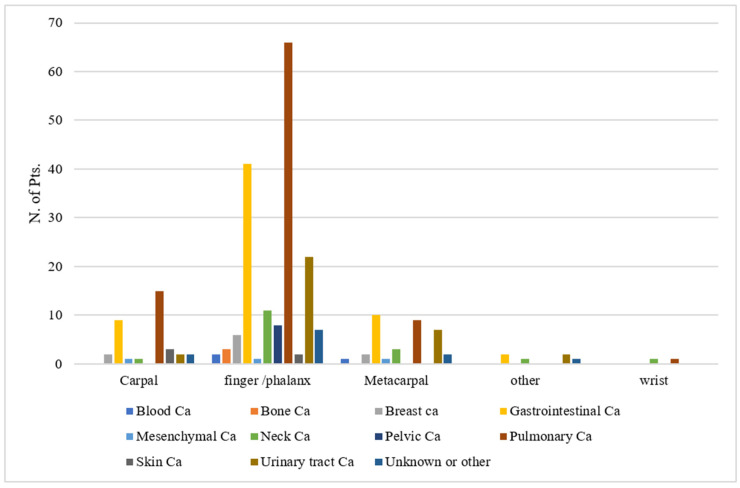
The distribution of acrometastases’ localization against the primary tumor site.

**Figure 3 medicina-57-00950-f003:**
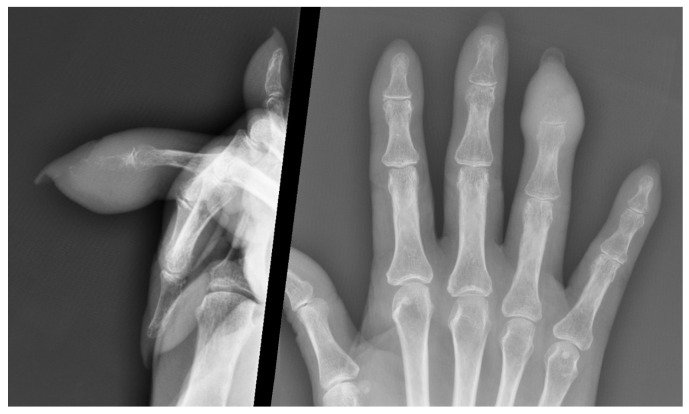
Hand X-ray showing thickening of the soft tissue of the right ring finger’s distal phalanx, with fracture of the distal phalanx and moderate loss of bone substance.

**Figure 4 medicina-57-00950-f004:**
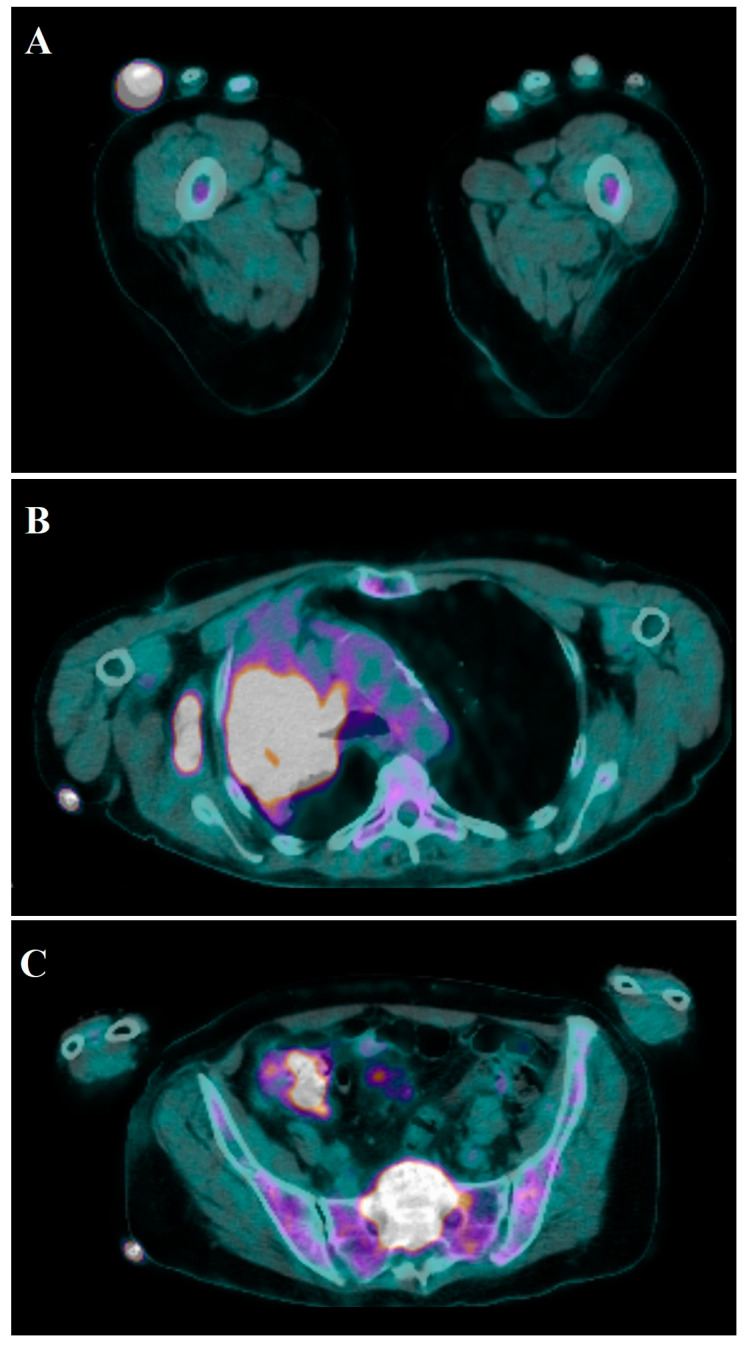
Total-body 18F-FDG PET-CT scans showing accumulation of tracer in the fourth finger of the right hand (SUV max 18.3) (**A**) and middle and upper lobes of the right lung (SUV max 15.8) associated with 18-F-FDG uptake in the lymph nodes of the right axilla, prevascular, right paratracheal, at the Barety space, at precarenal, subcarinal, and hilum of the right lung (SUV max 16.6) (**B**). Additionally, tracer uptake at T7, T8, T10, L3, sacrum, right acetabulum, and ipsilateral ischiopubic branch is shown (**C**)**.**

**Table 1 medicina-57-00950-t001:** The average interval from primary tumor diagnosis to acrometastasis (in months) according to the primary tumor type.

Primary Tumor Type	No. of Patients	Average Interval from Primary Tumor Diagnosis to Acrometastasis (Months)
Blood Ca	3	48
Bone Ca	3	44
Breast Ca	10	64
Gastrointestinal Ca	62	26.1
Mesenchymal Ca	3	2
Neck Ca	17	32.2
Pelvic Ca	8	6
Pulmonary Ca	91	13.4
Skin Ca	5	58
Urinary tract Ca	33	40.5
Other or unknown	12	27.3
Total	247	29.8

## Data Availability

Data sharing not applicable. No new data were created or analyzed in this study. Data sharing is not applicable to this article.

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
