# Peer review of "Acrometastases to the Hand: A Systematic Review"

_medicina, 2021, doi:10.3390/medicina57090950_

Round 1
Reviewer 1 Report
This is an extensive and well done review of the topic ACROMETASTASIS.- Congratulations.- Although this.. there are so much others "case reports", that are impossible to include.
I attach a review of 35 cases: (open acces)
Prognosis and treatment of acrometastases: Observational study of 35 cases treated in a single institution
V. Machado, M. San-Julian 10.1016/j.recote.2018.11.005 Rev Esp Cir Ortop Traumatol 2019;63:49-55
Author Response
We really thank R1 for her/his time spent to review this systematic review and for her/his appreciation.
We are truly sorry that we cannot include the clinical case series suggested (
Prognosis and treatment of acrometastases: Observational study of 35 cases treated in a single institution. V. Machado, M. San-Julian 10.1016/j.recote.2018.11.005 Rev Esp Cir Ortop Traumatol 2019;63:49-55) as in this specific case the acrometastases are found at the level of the lower limbs, while our systematic review is focused at the level of the hands and wrists.
We are grateful for the compliments received.
G E Umana
Reviewer 2 Report
This study is a systematic review of acrometastases to the hand. The authors analyzed available papers, including non-English ones.
The major flaws of this study:
- Figure 1 – in flow diagram 34 publications were excluded „with reasons” – what reasons authors have in mind
- furthermore described data extraction process contains numbers that don’t match to me with the flowchart presented and table 1 -
- table 1 seems to be too big for me to include into the original paper, it could be an appendix for download if possible. Most of the data is summarized in next chapters.
- In many papers included into the study there are gaps in interval time, metastases treatment and survival. The authors don’t comment on that further, which suggests that the number of papers taken into statistical analysis is the same along the parameters measured. There are also 227 studies in the table that doesn’t suite the number of papers included into the study,
- in the results authors mention other places of metastases – what are these places?
- discussion, lines 136-141 seems like a repeating the results section,
- the discussion seems to be short and includes sections that were not described as the aims of the study (imaging) and not included into results section (neither in the table 1),
- in the informed consent at the end, authors confirm to have written consent obtained from the patients, what patients do they have in mind?
My general suggestions are to clarity the issues and differences in the reviewed papers numbers, address the gaps and data deficiencies and improve discussion.
Author Response
We thank R2 for her/his time spent to review this systematic review. We really appreciate your suggestions that will certainly improve the manuscript.
Please find below our point-by-point reply to Your comments:
- - Figure 1 – in flow diagram 34 publications were excluded „with reasons” – what reasons authors have in mind
- reply:
- The reasons based on which the papers were excluded were the absence of one of more then one of the following points "study design, methodology, patient characteristics, and the following data points: authors, year of publication, patients’ age, sex, metastatic locations, primary tumor origin, interval time from primary tumor diagnosis to acrometastasis treatment options, and survival ". We indicated it already in the data analysis section. Fig 1, we will add in the Prisma diagram " Full text excluded with reasons 34: - incomplete data"
- - furthermore described data extraction process contains numbers that don’t match to me with the flowchart presented and table 1
- reply:
- we double checked table 1 and the results section and we did not found discrepancies. But probably we missed them, so please indicate them and we will promptly correct the manuscript
- - table 1 seems to be too big for me to include into the original paper, it could be an appendix for download if possible. Most of the data is summarized in next chapters.
- reply:
- we will remove table 1 and add it as supplementary material
- - in the results authors mention other places of metastases – what are these places?
- reply:
- i.e. lungs, skeleton, different sites than "acrometastasis, carpal metastasis, hand metastasis, finger metastasis, phalangeal metastasis, and wrist metastasis", please see methods section line 79
- There are also 227 studies in the table that doesn’t suite the number of papers included into the study
- reply
- table 1 includes 215 papers, not 227, please let us know how we can improve it further and we will do accordingly
- - In many papers included into the study there are gaps in interval time, metastases treatment and survival. The authors don’t comment on that further, which suggests that the number of papers taken into statistical analysis is the same along the parameters measured.
- reply:
- table 1 shows in the last 2 column "Metastases treatment" and "survival". Our goal was to show the type of treatment and the survival. So there is no gap, only the survival after the treatment. If we can improve or if we can explain this better please let us know and we will modify accordingly.
- - discussion, lines 136-141 seems like a repeating the results section,
- reply:
- we will remove those lines
- - the discussion seems to be short and includes sections that were not described as the aims of the study (imaging) and not included into results section (neither in the table 1)
- reply:
- the aim of the study is "to provide further insights into the management of these fragile patients" line 72. We hope that this could be considered as the aim of the study. Our goal was then to give all the information that was possible to obtain from the papers. Almost all case reports, this is the first systematic review. In our opinion the table 1 contains a large number of data and information and adding also information about pathophysiology or imaging could not be of benefit; we agree with you that table 1 can be uploaded as supplementary material. Finally, in the discussion section, we provided all the references from which we got the information about imaging, pathophysiology, patient population and management. Since most of the papers were case reports it was not possible, for example, to obtain "new" information about pathophysiology. So, we structured the discussion in subheadings, to better organize it. It was then not possible to statistically analyze imaging or pathophysiology in the results section, except for the management that we gathered also from our analysis and based on survival, treatment, interval time ect. The discussion section is about 1200 words, divided in 4 sections. We will try to improve it further.
- - in the informed consent at the end, authors confirm to have written consent obtained from the patients, what patients do they have in mind?
- reply:
- of course, we have in mind the images shown in fig 2 and 3, not all the patients presented in the systematic review. We could also provide a case illustration to show and emblematic patient, if required.
Round 2
Reviewer 2 Report
Authors commented on my remarks and improved the parts of the manuscript that were indicated in the review.
All remarks and doubts were clarified.